# COVID-19 Continues to Burden General Practitioners: Impact on Workload, Provision of Care, and Intention to Leave

**DOI:** 10.3390/healthcare11030320

**Published:** 2023-01-20

**Authors:** Anne Schrimpf, Markus Bleckwenn, Annett Braesigk

**Affiliations:** Department of General Practice, Faculty of Medicine, Leipzig University, 04109 Leipzig, Germany

**Keywords:** COVID-19, primary healthcare, patient care, vaccines, preventive health services, delivery of healthcare, patient care management, personnel turnover

## Abstract

General practitioners (GPs), already in a profession with a high workload, have been at the frontline of providing COVID-19-related healthcare in addition to routine care. Our study examined the impact of pandemic-related consultations and changes in practice organization on GPs’ current workload and provision of healthcare in summer 2021 (May 2021–July 2021) and early 2022 (January 2022–February 2022). In total, 143 German GPs participated in an online survey in the summer of 2021. Of these, 51 GPs participated in the follow-up survey in 2022. Most GPs perceived an increase in consultation frequency, consultation times, and workload since the pandemic outbreak. Increased consultation times were related to the reduced provision of medical care to other patients with chronic diseases. More SARS-CoV-2 vaccination consultations were associated with reduced home visits, acute consultation times, and cancer screenings. A quarter of GPs considered leaving their job. Pandemic-related bureaucracy, restricted access to therapy and rehabilitation services specialized on COVID-19, unreliable vaccine deliveries, mandatory telematics-infrastructure implementation, and frequent changes in official regulations were the main reasons reported for dissatisfaction. Our results provide insights into how the pandemic continues to burden GPs’ work routines and how better working conditions in times of high demand could be achieved in future pandemics.

## 1. Introduction

The coronavirus-19 disease (COVID-19) pandemic developed into a persistent burden for inpatient and outpatient healthcare and limited the capacities of the healthcare system to pursue day-to-day services [1,2,3]. In the outpatient sector, general practitioners (GPs) have been at the frontline of the COVID-19 pandemic. Considering that general practice is already a profession with a high workload [4,5], COVID-19 placed a number of additional burdens on GPs [6]. The management and treatment of patients with acute COVID-19 and/or with persistent symptoms after infection (defined as “long COVID” [7]) is in most cases provided by GPs and might develop—due to high prevalence [8] and treatment uncertainty [9]—into a long-term burden for GPs [10,11,12]. Further, GPs in Europe widely joined the national SARS-CoV-2 vaccination campaign [13], placing greater demands on them. Besides complying with changing hygiene measures and government regulations [14,15], the situation is further exacerbated by recurrent staff shortages due to sickness or quarantine [14] and increased psychosocial stress on the practice teams [16,17].

At the beginning of the pandemic, GPs largely restructured their work routines [18,19,20]. In Germany, planned procedures, preventive consultations, or continued care for chronically ill patients had to be canceled or postponed [21,22,23]. This trend has also been observed in other countries [24]. Some patients with chronic diseases might also have avoided (primary) healthcare facilities in the first months of the COVID-19 outbreak, to not be a burden [25] or out of fear of being infected with SARS-CoV-2 [26]. In this early phase, GPs in Germany and in other countries were confronted with a reduction of consultations [23,27] and, consequently, with economic concerns [28,29]. In addition, personal protective equipment was initially sparse, exposing healthcare workers to unsafe working conditions and health risks, which has been associated with concerns, anxiety, and burnout in physicians and nurses [30,31,32,33]. In line, frequent contact with patients with COVID-19 was related to higher scores in burnout and lower job satisfaction in health professionals [34,35,36].

After the first wave of the pandemic, in autumn 2020, suspended routine care slowly returned to pre-pandemic levels [27], leading to a double burden—COVID-19 and non-COVID-19-related healthcare—in the following waves of the pandemic [12,37]. Extensive administrative tasks, such as additional documentation, COVID-19 tests, phone counseling, and the implementation of hygiene measures, resulted in an additional workload [27,38,39,40]. Rapidly changing pandemic-related information was further identified as a stressor for healthcare providers [39,41]. Importantly, the provision of COVID-19-related medical care and administration largely superseded the healthcare services for patients with chronic diseases or routine care [10,42,43]. In this period, substantially lower quality of care in some patient groups, e.g., patients with diabetes, was observed [44]. Additionally, a previous review underlined an increase in psychological stress among GPs since the outbreak of the pandemic and their increased desire for leaving their job [17,45].

As most of these studies were conducted in the early phases of the pandemic, our study aimed to examine the ongoing impact of the pandemic, especially the double burden of providing healthcare to patients with COVID-19/long COOVID and routine care, on GPs’ work routines in mid-2021 (baseline) and at the beginning of 2022 (follow-up). These results might contribute to identifying potential areas for improvements in working conditions, and thus reduce further dissatisfaction, burnout, or withdrawals from practices. We invited German GPs to participate in our online-based questionnaire study, which was part of another study [9]. We hypothesized that COVID-19-related tasks increased GPs’ workload and disrupted routine care. At both time points, we specifically focused on GPs’ perceived changes in workload since the pandemic and investigated the impact of consultations related to COVID-19 on workload. For this, we also assessed the current number of patients with long COVID treated in the practice. We further examined the impact of the pandemic on GPs’ perceived quality of provided routine healthcare and perceived ability to provide healthcare to patients with acute COVID-19 and long COVID. Potential changes in these variables between the two time points were analyzed. At baseline in 2021, we additionally explored potential challenges related to the provision of healthcare to patients with COVID-19 and GPs’ needs for information and support with respect to long COVID. In addition, the frequency of consultations of patients with long COVID was examined. At follow-up in 2022, we assessed the current main reasons for consultations in the practice as well as which healthcare services were currently reduced to determine potential changes in favor of COVID-19-related healthcare. Further, the number of administered SARS-CoV-2 vaccine doses was assessed, as large vaccine centers have been closed at the time of the follow-up survey in Germany [46]. Lastly, GPs’ intention to leave practice and perceived main burdens related to practice organization during the pandemic were examined.

## 2. Methods

### 2.1. Recruitment Procedure

The data from the online surveys were collected in the Free State of Saxony, Germany, between May 2021 and July 2021 (defined as baseline 2021 in the following text). A follow-up survey was sent between January 2022 and February 2022 to all GPs who participated in the first assessment (defined as follow-up 2022 in the following text). The recruitment process of the first assessment has been described in more detail by Schrimpf et al. [9].

GPs were selected by the availability of an email address, on file with the Association of Statutory Health Insurance Physicians Saxony (Kassenärztliche Vereinigung Sachsen), and they were invited by email to voluntarily participate in this online survey. In May 2021, the first invitation email was sent to 1444 GPs. The email contained information about the purpose of the survey, data handling, and a link to the online survey. The access to the survey was generated by a TAN (transaction authentication number) to ensure that each GP participated only once. In June and July 2021, reminders containing the same information were sent to non-responders (following the recommendations of Edwards et al., [47]). In January 2022, an invitation to a follow-up survey was sent to all GPs who participated in the first survey, identified by their email addresses. Two reminders at the end of January and in mid-February 2022 were sent to non-responders (see recruitment process in Figure 1).

### 2.2. Survey

The questionnaires were self-developed in the Department of General Practice of Leipzig University by an interdisciplinary research team (medical scientists, biochemist, and GPs) in a multi-stage revision process. The development process was complemented by an extensive literature search aimed at identifying relevant factors for barriers in ambulatory care related to patients with COVID-19 or long COVID. The final versions of both the initial and the follow-up survey can be found in Appendix A. For the web-based evaluation, the software LimeSurvey (http://www.limesurvey.org/) *was used*, hosted on a secure server of the Leipzig University Computer Center. The completion of the first online survey took approximately 15 min and the follow-up survey took approximately 10 min.

Participating GPs were asked in both surveys to click the “I agree” button on the online informed consent form. Then, the survey started and comprised of the following topics: (1) demographics (e.g., age, sex, practice information), (2) patients with long COVID in the practice (e.g., frequency of consultations, difficulties during treatment, need for information or support), and (3) impact of the COVID-19 pandemic on practice and patient care. The response formats were multiple choice answers, rating scales (1–10), and free text entries.

Prior to implementation, the questionnaires underwent a think-aloud pre-testing [48] aimed at identifying problems or misunderstandings related to each item. The provisional questionnaires were filled out by five GPs, who were instructed to think aloud while answering each item and report every spontaneous thought. After completing the questionnaires, the GPs were briefly interviewed about general issues with the questionnaires (e.g., length, structure, and general comprehensibility). After pre-testing, the provisional questionnaires were adjusted and further developed.

### 2.3. Coding of Free Text Entries

Participating GPs were asked to indicate in free text fields if needed, additional problems, and needs related to the treatment of COVID patients as well as reasons for wanting to quit their jobs. Additional general comments and wishes could be entered at the end of the survey. Free text entries were independently coded by two authors of this study (AS, AB). Categories were derived inductively during the coding, either indicating a major category or a subcategory. The assignments were compared and differences in coding were discussed until an inter-coder agreement was reached for each discrepancy.

### 2.4. Statistical Analyses

All statistical analyses were carried out using IBM SPSS Statistics 27 (Armonk, NY, USA) with a two-sided α-level of 0.05. For descriptive statistics, missing values in single variables were considered by presenting frequencies as % (n/n_valid_). Continuous variables were presented as mean (M) ± standard deviation (SD).

Differences between categorical variables were analyzed using chi-square or Fisher’s exact tests, respectively. Fisher’s exact tests were used when 20% of the cells in the contingency table had expected frequencies < 5. We specifically analyzed differences between GP practices with and without a special focus on COVID-19 in the following categorical variables: “sex” (male, female), “specialization general medicine” (yes, no), “specialization internal medicine” (yes, no), “specialization others” (yes, no), “practice structure” (single practice, practice sharing, joint practice, medical center), “Number of cases per quarter” (≤700, 701–1000, 1001–1500, ≥1501, no answer), “catchment area city” (yes, no), “catchment area small town” (yes, no), and “catchment area rural” (yes, no).

Univariate analyses of variance (ANOVAs) were used for analyzing differences in continuous variables between groups. Repeated measures ANOVAs were applied with the within-subject factor “time” (baseline 2021, follow-up 2022) for the metric variables “numbers of patients with long COVID”, “Being able to address medical and/or psychological needs of patients with COVID-19”, and “Perceived limited provision of satisfactory medical care to other patients with chronic diseases” to measure differences between the two time points. Estimated effect sizes were reported using partial eta squared (η_p_^2^). For all univariate and repeated measures ANOVAs indicating a significant main effect, least significant differences tests were utilized to determine the origin and direction of the effect, in which case we report M ± SD.

Further, a two-sided bivariate correlation was calculated to analyze the association of two continuous variables (number of patients with long COVID treated in the practice and the satisfactory provision of medical care to other patients with chronic diseases).

## 3. Results

### 3.1. Sample Characteristics

Of the 186 GPs who participated in this study (13% total response rate), 45 GPs left the survey incomplete after the first page. In total, 143 GPs were included in the analyses at baseline. Of these participants, 51 completed the follow-up survey in January or February 2022. We further differentiated practices stated to have or not to have a special focus on COVID-19, indicating that especially practices with more health insurance approved physicians specialized in COVID-19 management (Table 1). Practices with a focus on COVID-19 treated on average more patients with long COVID (symptoms lasting between 4 to 12 weeks: *M* = 20.1, *SD* = 14.3; symptoms lasting more than 12 weeks: *M* = 9.8, *SD* = 8.1) than standard practices (symptoms lasting between 4 to 12 weeks: *M* = 11.2, *SD* = 10.9, *F*(1, 129) = 5.451, *p* = 0.021, η_p_^2^ = 0.041; symptoms lasting more than 12 weeks: *M* = 4.5, *SD* = 5.9, *F*(1, 128) = 6.345, *p* = 0.013, η_p_^2^ = 0.047). Additional percentages, means, and standard deviations for GP sample characteristics and practice information can be found in Table 1. A direct comparison of GP sample characteristics between baseline 2021 and follow-up 2022 can be found in Appendix A.

### 3.2. Workload since the Outbreak of the Pandemic

Information on changes in workload, economic situation of the practices, frequencies of consultations, and time requirements for consultations for both time points can be found in Table 2. In general, most GPs perceived an increase in the frequency of patient visits, consultation times, and workload since the outbreak of the pandemic. Between baseline 2021 and follow-up 2022, the frequency of patient visits continued to increase. The economic situation of the practices only worsened in a minority of practices.

### 3.3. Consultations Related to COVID-19 in GP Practices

**Comparison between the two surveys:** The number of patients with long COVID remained stable from baseline 2021 (*M* = 11.9 patients with long COVID symptoms lasting between 4 to 12 weeks, *M* = 5.9 patients with long COVID symptoms lasting more than 12 weeks) to follow-up 2022 (*M* = 10 patients with long COVID symptoms lasting between 4 to 12 weeks, *M* = 6.8 patients with long COVID symptoms lasting more than 12 weeks) in GP practices (Table 3). Repeated measures ANOVAs revealed no significant differences between the two time points (patients with long COVID symptoms lasting between 4 to 12 weeks: *F*(1, 48) = 0.081, *p* = 0.777, η_p_^2^ = 0.002; patients with long COVID symptoms lasting more than 12 weeks: *F*(1, 49) = 1.687, *p* = 0.200, η_p_^2^ = 0.033).

**Baseline 2021:** The majority of GPs (51.9%) reported treating patients with acute COVID-19 every week in their practice. Similarly, most GPs (44.3%) stated they treated patients with long COVID symptoms lasting between 4 to 12 weeks at least once a week in their practice. In contrast, patients with long COVID symptoms lasting more than 12 weeks were largely treated at least once a month (44.6%) in GP practices (Table 3).

**Follow-up 2022:** GPs were asked to estimate how many out of 100 patients in their practice are currently visiting for pre-defined counseling needs. Importantly, GPs reported that, in total, 29.4 out of 100 patients are currently visiting the practice because of COVID-19/long COVID or SARS-CoV-2 vaccination issues. In addition, GPs were asked to indicate which medical services can currently only be offered at a reduced capacity. Self-payer services, preventive health check-ups, and preventive cancer screenings have been offered less by 66%, 58%, and 44% of respondents, respectively. Further details for the variables assessed can be found in Table 3.

### 3.4. Provision of Care for Patients with Long COVID: Identified Problems and Need for Support

**Comparison between the two surveys:** GPs were asked at both time points whether they were able to address the medical and/or psychological needs of patients with acute COVID-19 or long COVID during consultations on a scale from 1 = “not able” to 10 = “fully able”. The ratings did not change between baseline 2021 and follow-up 2022 (*F*(1, 48) = 0.096, *p* = 0.758, η_p_^2^ = 0.002; Table 4).

**Baseline 2021:** GPs were asked at baseline to identify current problems related to the treatment of patients with long COVID and the GPs’ need for information. The long course of the disease (78.9%) was rated as the main problem during the treatment of patients with long COVID. The introduction of national guidelines on long COVID (63.9%) was identified as the main need for support of GPs. All results can be found in Table 4. GPs were additionally able to make individual comments on both questions. The lack of and need for specialists and facilities to treat patients with long COVID were mentioned by most respondents. The results of these free-text answers can be found in Appendix A.

GPs who reported a worsening of the economic situation of their practice since the outbreak of the pandemic rated their ability to address COVID patients’ needs as lower (*M* = 5.5, *SD* = 2.0) compared to GPs who reported an unchanged (*M* = 6.7, *SD* = 1.9) or improved economic situation (*M* = 7.4, *SD* = 1.7) of their practice (*F*(2, 127) = 4.985, *p* = 0.008, η_p_^2^ = 0.073; Figure 2A). In addition, GPs who stated that long COVID is difficult to diagnose rated their ability to address their patients’ needs as lower (*M* = 6.3, *SD* = 1.9) compared to GPs who did not report diagnostic difficulties (*M* = 7.0, *SD* = 2.0; *F*(1, 130) = 4.468, *p* = 0.036, η_p_^2^ = 0.033). Further, GPs who stated that unspecific symptoms of long COVID are problematic rated their ability to address their patients’ needs as lower (*M* = 6.5, *SD* = 2.0) compared to GPs who did not report problems with unspecific symptoms (*M* = 7.4, *SD* = 1.9; *F*(1, 130) = 4.407, *p* = 0.038, η_p_^2^ = 0.033).

### 3.5. Provision of Care for Other Patients since the Outbreak of the Pandemic

**Comparison between the two surveys:** GPs were asked whether they felt that the pandemic limited the provision of satisfactory medical care to other patients with chronic diseases on a scale from 1 = “fully limited” to 10 = “not limited”. The ratings changed between baseline 2021 and follow-up 2022 (*F*(1, 48) = 11.287, *p* = 0.002, η_p_^2^ = 0.190; Table 4), indicating a perceived worsening over the two time points.

**Baseline 2021:** GPs felt more limited in the satisfactory provision of medical care to other patients with chronic diseases the more patients with long COVID were currently treated in the practices (*r*(129) = 0.18, *p* = 0.039). Further, GPs who stated an increase in time for patient consultations felt more limited in the satisfactory provision of medical care to other patients with chronic diseases (*M* = 5.8, *SD* = 2.5) compared to GPs who reported an unchanged (*M* = 7.1, *SD* = 2.1) or decreased (*M* = 7.4, *SD* = 2.3) time investment (*F*(2, 128) = 4.716, *p* = 0.011, η_p_^2^ = 0.069; Figure 3A).

**Follow-up 2022:** GPs were asked to indicate which healthcare services were currently less likely to be offered. The results showed that especially preventive cancer screenings, preventive health check-ups, and self-payer services were currently reduced (Table 4). In addition, GPs who reported currently offering fewer acute consultation times than usual administered more SARS-CoV-2 vaccinations per week (*M* = 124.4, *SD* = 95.1) than GPs who did not change their acute consultation services (*M* = 63.0, *SD* = 63.9; *F*(1, 46) = 5.578, *p* = 0.022, η_p_^2^ = 0.108). In the same line, GPs who reported currently offering fewer home visits than usual administered more SARS-CoV-2 vaccinations per week (*M* = 118.7, *SD* = 109.7) than GPs who did not change their home visit services (*M* = 59.8, *SD* = 51.2; *F*(1, 46) = 6.399, *p* = 0.015, η_p_^2^ = 0.122; Figure 3B). Further, the percentage of patients currently visiting for SARS-CoV-2 vaccinations was related to GPs’ ability to provide preventive cancer screenings, showing that GPs who currently reduced their cancer screening services had a higher proportion of SARS-CoV-2 vaccination consultations in their practice (*M* = 18.8, *SD* = 11.5) compared to GPs with no changes in cancer screening services (*M* = 12.4, *SD* = 7.9; *F*(1, 46) = 5.178, *p* = 0.028, η_p_^2^ = 0.101). No relationship between SARS-CoV-2 vaccination consultations and other services has been found.

### 3.6. Intention to Leave

At follow-up, GPs were able to indicate if they considered quitting their job in the last 12 months, which 26.5% affirmed (Table 2). Age did not differ between GPs who affirmed (*M* = 47.7, *SD* = 9.2) and those who did not affirm having considered quitting (*M* = 49.5, *SD* = 9.2). We further found that GPs who considered quitting their job also currently treated more patients with long COVID (patients with symptoms lasting between 4 to 12 weeks: *M* = 15.7, *SD* = 15.1; patients with symptoms lasting more than 12 weeks: *M* = 10.1, *SD* = 12.4) compared to GPs who did not consider quitting (patients with symptoms lasting between 4 to 12 weeks: *M* = 7.0, *SD* = 5.5, *F*(1, 45) = 8.603, *p* = 0.005, η_p_^2^ = 0.160; patients with symptoms lasting more than 12 weeks: *M* = 4.3, *SD* = 3.5, *F*(1, 46) = 6.394, *p* = 0.015, η_p_^2^ = 0.122; Figure 2B).

GPs were able to indicate in free text fields reasons for considering leaving their job. We identified the following main reasons: increase in workload and administrative tasks, demanding patients, as well as the handling of the pandemic by politicians, health authorities, and media. All results of these free text answers can be found in Appendix A.

### 3.7. Additional Comments and Wishes

**Baseline 2021:** At the end of the baseline survey, participants were asked to provide additional comments regarding long COVID, and 33 GPs filled in the free text field. A summary of statements can be found in Appendix A. The majority of statements were related to the current treatment of long COVID and GPs’ individual observations regarding the long course of the disease. Many GPs wished for better therapy and rehabilitation options for their patients—also with respect to psychotherapy—and described the current possibilities as insufficient:


*“It is difficult to get a rehabilitation place (which is also time-consuming and help is needed). The same applies to initiating psychological co-treatment. The health insurance companies do not support me as a doctor and my patients (e.g., I went through depressing written disputes about quarantine/AU [certificate of incapacity for work]).”*

*(female GP, 49 years old)*



*“Direct and timely access to rehabilitation and specialist care must be organized! The best way is via a central coordination office. It is essential to set up a quota for psychotherapy for these patients!”*

*(female GP, 42 years old)*


In addition, GPs reported the observation of strong psychological comorbidity in their patients with long COVID. Whereas some GPs see an increase in psychosomatic symptoms after infection with SARS-CoV-2, others attribute these symptoms to a pre-existing psychological condition:


*“In my patients I see predominantly psychological impairments, especially an increase in anxiety/neurotic symptoms accompanied by physical and cognitive stress insufficiency. It is difficult to differentiate whether the physical limitations are a consequence of the psychological impairments.”*

*(female GP, 45 years old)*



*“I am concerned that this disease is drifting more into the psychosomatic domain. Apart from a long feeling of illness, there is no tangible value and no recovery criterion except for the patient’s subjective statements.“*

*(male GP, 59 years old)*



*“More than genuine “post/long COVID symptoms”, we observe an aggravation of already psychologically pre-altered patients in connection with COVID-19 without objectifiable pathological organic findings.”*

*(male GP, 39 years old)*


Only a few GPs mentioned perceived issues with media coverage, research, and politics and their influence on the practice and patients:


*“The extensive “nocebo education” provided by the media and the constant change of information are counterproductive for physical and psychological convalescence.”*

*(male GP, 39 years old)*



*“The state has failed and these polls are far too late. Last year’s discussions [2020] were a disgrace to the academy. Germany is stuck in the Middle Ages when it comes to communication between the university and the front.”*

*(male GP, 35 years old)*


**Follow-up 2022:** At the end of the follow-up survey, participants were asked about their wishes for improvements, and 38 GPs filled in the free text field. A summary of statements can be found in Table 5. The majority of statements were related to politics and regulations, indicating that GPs wished for a substantial reduction in bureaucracy and administrative work:


*“At the moment, I am only 50% GP and 50% practices organizer. Bureaucracy is not diminishing, since Corona, it massively increased (through constant change of billing codes, diagnosis codes and combinations, official orders).”*

*(male GP, 54 years old)*


Further, many GPs wished for more reliable policy announcements and a reduction of political short-notice decisions during the pandemic. Along the same line, many GPs wished for reliable vaccine dose orders to organize their practice and for SARS-CoV-2 vaccine offers outside the practice:


*“Planning security. A reasonable, comprehensible, and not constantly changing strategy in pandemic control and vaccine supply. Relief through sufficient vaccination services outside the practice. We can well and safely secure the infection event and the outpatient care of patients suffering from COVID-19 if we are not responsible for the quarantine regulations and we also receive the ordered vaccine. Compulsory vaccination of staff in our facilities will lead to staff shortages and I worry that, then at the latest, we will only be able to provide minimal patient care and the quality of care can no longer be guaranteed due to overwork of the remaining staff.”*

*(female GP, 36 years old)*


It was further mentioned that the mandatory implementation of telematics infrastructure (enabling an electronic patient file, electronic prescriptions, and electronic certificate of incapacity for work) during an already pandemic-related high workload additionally burdened GP practices:


*“The IT innovations are justified, plausible, and at some point perhaps also facilitating/helpful. Currently, however, these things represent an additional burden! It would therefore be helpful to postpone them or to implement simplified processes!”*

*(male GP, 43 years old)*



*“It would help to be released from the burdensome and largely pointless expansion of digitization applications. I am not an opponent of digitization, but the currently planned measures such as e-prescription, e-AU [certificate of incapacity for work], and e-PA [patient file] predominantly cost time, money, non-existent mental reserves without visible practicability for the general public.”*

*(female GP, 46 years old)*


**Table 5 healthcare-11-00320-t005:** GPs’ wishes for improvements (*n* = 38) at follow-up 2022: content analysis of free text answers.

Major Category	Subcategory	*n* *	% **
General public and politics		30	78.9
	Reduction of bureaucracy	12	31.6
	Reduction of short-notice decisions/changes	9	23.7
	Reliable policy announcements	7	18.4
	Improvement of remuneration and budgeting	6	15.8
	Scientifically transparent recommendations	4	10.5
	Easy-to-understand regulations regarding quarantine and masks	4	10.5
	Improved collaboration with health authorities	3	7.9
	Consistent regulations for quarantine	2	5.3
	Unburdening of the reporting system for infectious diseases	2	5.3
	Coverage of non-medical COVID-19 counseling services by (health) authorities	2	5.3
	Anticipatory and responsible media communication	2	5.3
	GPs included in political committees and advisory boards	2	5.3
	Organizational system for scarce resources	1	2.6
	Actions against vaccination opponents and misinformation	1	2.6
	Reduction of mandatory health insurance services	1	2.6
	Protection of the health system from investors and shareholders	1	2.6
Everyday practice		22	57.9
	More time for digital implementations	5	13.2
	No mandatory digitalization	5	13.2
	Increased time for patients (with chronic conditions)	3	7.9
	Improved availability of psychotherapists	3	7.9
	Preservation of therapeutic freedom	2	5.3
	Working without masks and/or tests	2	5.3
	Short hand-outs with information (on treatment of patients with COVID-19/long COVID, COVID-19 testing)	2	5.3
	Local meetings with other physicians	1	2.6
	Increased digitalization (e-vaccination card, e-prescription)	1	2.6
	Improved appointment options in long COVID outpatient clinics	1	2.6
Vaccination		18	47.4
	Reliability of vaccine dose orders	10	26.3
	Constant offers for testing and vaccination outside the practice (test and vaccine centers)	5	13.2
	No compulsory vaccination	5	13.2
	Consistent regulations for COVID-19 vaccinations	2	5.3
	SARS-CoV-2 vaccines in single vials	1	2.6
	Compulsory vaccination	1	2.6
Staff and colleagues		13	34.2
	Improvement of staffing situation	4	10.5
	Financial and societal upgrading of the medical staff	4	10.5
	Protection of staff from insults, threats, and misinformation	2	5.3
	Better medical cooperation in outpatient and inpatient areas	2	5.3
	Promotion of training and professional profile of physician assistants	2	5.3
	Physician assistant for administrative tasks	1	2.6
	Ensured provision of child care	1	2.6
	Recruitment of medical staff through health insurances	1	2.6
	Better work-life balance	1	2.6
	Possibility to delegate more non-medical tasks to staff	1	2.6
	Clear regulations regarding unvaccinated staff	1	2.6

*n* * = statements in this category, ** % = percentage of GPs who stated an insight from this category.

## 4. Discussion

The present study investigated the impact of the COVID-19 pandemic on GPs’ workload, quality of patient care provision, intention to leave, and working conditions in 143 German general practices in 2021 and followed up with 51 of those in 2022. Most GPs perceived an increase in the frequency of patient visits, consultation times, and workload since the outbreak of the pandemic. At baseline 2021, increases in consultation times were related to perceived limitations in the satisfactory provision of medical care to other patients with chronic diseases. At follow-up in 2022, an increase in the number of SARS-CoV-2 vaccination consultations conducted was associated with reduced care services in the practices, such as home visits, acute consultation times, or cancer screenings. Better access to therapy and rehabilitation for patients with long COVID, especially psychotherapy, was identified as the main need of GPs. We further found that a quarter of GPs considered leaving their job, which was related to the current number of patients with long COVID in the respective practices. Increased administrative tasks, unreliable vaccine dose deliveries, simultaneous introduction of telematics-infrastructure implementation, as well as the handling of the pandemic by politicians, health authorities, and media were identified as reasons for dissatisfaction. Our results provide insights into how the pandemic continued to burden GPs and their work routines between 2021 and 2022.

### 4.1. Pandemic’s Influence on Workload

Most GPs in our study perceived an increase in workload, consultation times, and frequency of consultations since the outbreak of the pandemic, the latter further amplified between 2021 and 2022. Our results are in line with both qualitative research [12] and survey results [49] from Germany and other countries [45,50]. In addition, a recent study using medical record data reported that the average number of consultations in GP practices in the summer of 2021 increased by 18% as compared to a comparable period in 2019 [51]. Longer consultation times might have resulted from increased patient requests [37] and from the double burden of providing regular care and COVID-19-related care [12]. Especially communicating COVID-19-related information, e.g., about vaccines, potential symptoms, or testing, has been found to be common in GP practices [27,38,40].

### 4.2. Consultations Related to COVID-19 and Other Services

Compared to the baseline survey [9], we found that the number of patients with long COVID in GP practices remained stable from 2021 to 2022. At follow-up in 2022, we further found that almost one-third of all consultation issues were related to COVID-19, including patients with acute/long COVID-19 and SARS-CoV-2 vaccinations, indicating that these healthcare services came at the expense of other essential services. Our results showed that especially care of chronically ill patients, preventive cancer screenings and health check-ups, as well as self-payer services, were reduced by many GPs. In line with our findings, previous research reported a substantial decrease in new cancer diagnoses, especially in GP practices, since the outbreak of the pandemic [22,52] as well as a disruption of chronic disease management [53]. The pandemic’s negative impact on primary care for non-COVID patients due to shifted resources has also been discussed in other studies [12,54]. Although medical record data showed a general increase in GP consultations since the pandemic, diagnoses of new diseases dropped by 6% between these two periods [51]. An additional physician questionnaire revealed substantial reductions in home visits and opening hours as well as suspended check-ups and delayed consultations for high-risk patients by physicians in Germany during the pandemic, further indicating a shift in healthcare services [51].

### 4.3. Pandemic’s Influence on the Provision of Patient Care

Shifted resources to pandemic-related care might come at the expense of chronically ill patients. Our results support this assumption: at baseline in 2021, GPs rated the satisfactory provision of medical care to patients with chronic diseases as more limited, the more patients with long COVID were currently treated in their practices and the more they perceived an increase in time for patient consultations since the outbreak of the pandemic. Importantly, the perceived limited provision of satisfactory medical care to patients with chronic diseases further amplified between 2021 and 2022, indicating an ongoing burden of the pandemic on patient care. Although patient consultations dropped during the first wave of the COVID-19 pandemic [23,27], during the following waves primary care was overwhelmed by the double burden of managing pandemic-related care and routine care with the same pre-pandemic resources [12,54,55], potentially leaving patients with chronic conditions underserved [22,43,44].

In addition, in 2021, GPs whose practices had a worsening economic situation since the outbreak of the pandemic rated their ability to address COVID patients’ needs as lower compared to GPs with an unchanged or improved economic situation. This finding indicates that adjustments to the billing system in the context of changes in patients’ needs and consultation reasons since the pandemic might be necessary and at the same time beneficial for patient care.

At follow-up in 2022, especially the administration of SARS-CoV-2 vaccinations was associated with reduced offers of medical services, such as acute consultation times, home visits, or preventive cancer screenings. Since April 2021, GP practices were allowed to join the national vaccination campaign in Germany [13,56]. The proportion of SARS-CoV-2 vaccines administered in GP practices in January and February 2022 was significantly higher than at the beginning of the vaccine campaign in April 2021 [57], potentially influenced by both the COVID-19 winter surge and the dismantling of high-capacity but expensive mass vaccination centers since September 2021 [46]. Our results suggest that the reduction of mass vaccination sites, such as vaccine centers, increased the burden on GP practices and came at the expense of other essential primary care services. We argue that external vaccination offers, especially in the upcoming COVID-19 waves, might be beneficial to relieve the burden on GPs and ensure the delivery of routine primary care services.

### 4.4. Intention to Leave

We asked GPs at follow-up in 2022 whether they were considering leaving their job in the last 12 months, which one-quarter of the respondents affirmed. Our results resonate with a previous study from the US, showing that 24% of participating physicians indicated a moderate to high likelihood to leave their current practice within the next two years [58]. Especially increases in workload and COVID-19-related stress have been found to be associated with the intention to leave [58], which was also next to patients’ attitudes as stated in free text fields by GPs in our study. In addition, the age of the respondents was not a determinant for considering leaving practice, which was also found in a previous study with German GPs prior to the pandemic [59]. We further found that a greater number of patients with long COVID treated in GP practices was related to GPs considering quitting and might contribute to an increased burden. In line, previous studies found that frequent contact with patients with COVID-19 was related to higher scores in burnout and lower job satisfaction in health professionals [34,35,36]. In addition, qualitative data showed that the double burden of maintaining regular healthcare and COVID-19-related healthcare was perceived to be exhausting [12,37,42]. Generally, studies showed that the pandemic placed high psychological burdens on healthcare workers [17,60], which potentially accelerated during the course of the pandemic [49]. Our results are of concern, as a previous study showed that GPs’ intention to leave patient care was a predictor of actually leaving their job [61]. The additional pandemic-related workload might therefore be a catalyst for GPs’ intention to leave.

### 4.5. Perceived Burdens and Need for Support

In free text fields, GPs expressed at baseline in 2021 the urgent need for specialists, outpatient and rehabilitation clinics, and psychotherapies for patients with acute COVID-19 and long COVID. The expansion of care services related to COVID-19 as well as structured concepts was also requested by German patients [62]. Our results emphasize the benefits of the previously described interdisciplinary, multi-sectoral, and interprofessional approach to the management of patients with acute COVID-19 and long COVID [62,63,64], it being able to meet the varying needs of affected patients.

At follow-up in 2022, especially bureaucracy and administrative work, which increased during the pandemic, were perceived as burdens and were also reported elsewhere [27,65]. Inflated bureaucracy and over-regulation have been discussed as shortcomings of the governments’ pandemic regulation attempts [66] and should be further evaluated and revisited to reduce the workload of healthcare providers. Reliable policy announcements were additionally mentioned by GPs in our study as being needed for better planning security. In line, constant changes in official information or announcements have been found to also generate stress, confusion, and workload in healthcare providers of other countries [27,37,39,41,67]. Reduction of information in terms of frequency and quantity might therefore be beneficial in future pandemics to increase GPs’ and patients’ adherence to these announcements and regulations. Some GPs also wished for constant vaccination opportunities outside the practice and for reliable vaccine dose deliveries. In Germany, SARS-CoV-2 vaccinations have been found to be associated with high efforts and administrative work and to be insufficiently remunerated in GP practices [68]. These time expenditures increased the GPs’ workload and may have resulted in reduced capacities for other healthcare services. In addition, GPs could order vaccines only in limited quantities and vaccine doses were delivered depending on availability, further increasing GPs’ planning insecurity and workload. Lastly, GPs perceived the implementation of telematics infrastructure at the time of the pandemic as an additional burden. Coincidentally, the obligation to establish certain telematics infrastructure functions in German medical practices was introduced during the COVID-19 pandemic [69], which came with initial technical difficulties, such as malfunction or compatibility issues. As has been shown elsewhere [70,71], GPs in our study stated a general affinity for digitalization and acknowledged the benefits. However, due to initial problems, implementation was time-consuming, in parallel with an already increased workload caused by the pandemic. In particular, the perception of time savings through digitalization was mentioned by physicians in other studies as a facilitator for adopting digitalization [70,71]. In sum, our results indicate that increases in workload for GPs during the pandemic had multiple drivers, including bureaucracy related to pandemic regulation, frequent changes in official information and legislation, being responsible for the main coverage of the national SARS-CoV-2 vaccination strategy, and the simultaneous requirement to implement new, not fully developed, telematic infrastructures.

### 4.6. Limitations

Our study has limitations. First, given the nature of the study and the sample size, a selection bias might have occurred. All answers were self-reports and might be imprecise due to subjective perceptions. Some of the reported results might be influenced by other factors than pandemic-related work changes. In addition, we did not conduct a power analysis, wherefore the data are to be considered exploratory and cannot be generalized. Second, our questionnaire is not a valid scale as we did not develop and assess several items measuring a construct related to workload, provision of patient care, or job satisfaction, but rather investigated single-item responses. Single-item responses were chosen over scales to reduce the length and monotony of the questionnaire and, hence, increase willingness to participate in a population with time constraints. However, studies showed that single-item responses might be as reliable as multiple-item scales, especially for less complex constructs (e.g., [72,73,74]). Third, due to a cross-sectional study design, we do not have data on GPs’ status before the pandemic and can only depict subjective perceptions of changes since the pandemic. Lastly, the study was conducted in one federal state in Germany. Differences (e.g., in pandemic regulations, vaccine supply, or case incidences of COVID-19) between federal states in Germany as well as between European countries limit the generalizability or comparability of our findings.

### 4.7. Implications

Our research contributes to a better understanding of the ongoing impact of the COVID-19 pandemic on the provision of primary healthcare and GPs’ satisfaction with their working conditions. As GPs’ workload was already high in pre-pandemic periods [4], the current conditions have been described as unsustainable [6]. Our results might therefore have some general implications. Considering the average age of the GPs, demographic changes, and the expected decrease in treatment capacities in the future, the COVID-19 pandemic might act like a magnifier of the upcoming problems and distribution battles, such as prevention and screening vs. acute treatment as well as the lack of referral of patients to specialists or clinics. The GP as the gatekeeper would become the universal treatment provider. Further, the profession of general practice is already suffering from a lack of attractiveness for young doctors [75,76] and is perceived to have a disproportionately high amount of administrative tasks and comparatively low income [77]. The extraordinarily high workload and increased administrative tasks experienced by GPs during the pandemic might further deter young graduates to pursue a career in general practice.

To reduce further dissatisfaction, burnout, or even withdrawals from practices, our results highlight the following potential areas for improvement (Box 1).

Box 1Potential improvements of GPs’ working condition. 
(1)The primary care sector carries the main quantitative burden of care for patients with acute COVID-19 and long COVID as well as SARS-CoV-2 vaccinations. GPs would therefore potentially benefit from extended treatment options, including additional external vaccination offers and referral options for patients with acute COVID-19 or long COVID to specialists or rehabilitation to increase time for non-COVID-19-related healthcare. Information and updates on these additional local specialized care offers should be easily available for GPs. In addition, multidisciplinary teams and the possibility to allocate tasks to practice nurses might further reduce GPs’ burden.(2)In the course of this pandemic or in future pandemics, GPs can be supported by revisiting, suspending, or outsourcing pandemic-related documentational and administrative work.(3)Changes in official regulations and legislation should be reduced to a minimum to increase compliance.(4)Practices should be able to postpone the implementation of obligatory changes in practices’ structures during times of extraordinarily high workload, such as telematics infrastructures. These implementations should also be fully developed and should not cause additional workload.(5)Finally, feeling valued for the daily responsibilities as well as financial incentives could further increase satisfaction for staff working in general practices and might compensate for the higher workload, as has also been reported previously [58].


## 5. Conclusions

GPs are at the forefront of providing COVID-19-related healthcare in addition to routine care. We confirm that the pandemic continues to aggravate GPs’ working conditions and affect other essential healthcare services. We found evidence that, without political mitigation measures, the pandemic might accelerate GPs’ intention to leave the practice. Especially reductions in bureaucracy, the provision of additional vaccination sites or referral options for patients with acute COVID-19 or long COVID, and the option to postpone telematic infrastructure implementations in times of increased workload might contribute to the alleviation of GPs’ current and future working conditions. Our findings provide insights into how future pandemics could be handled to achieve better primary care working conditions in times of high demand.

## Figures and Tables

**Figure 1 healthcare-11-00320-f001:**
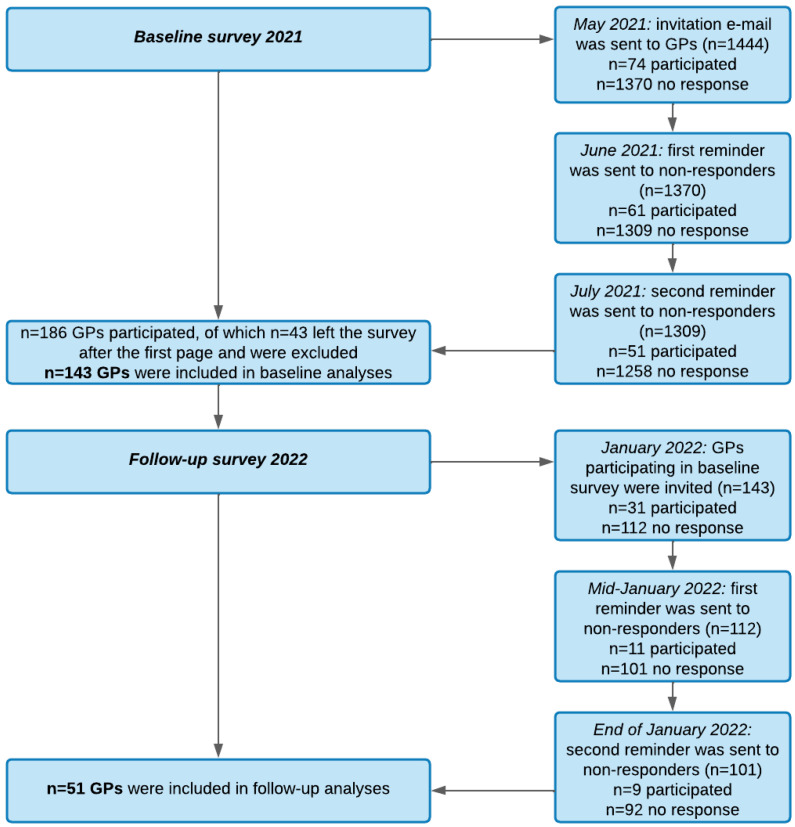
Flowchart of recruitment process.

**Figure 2 healthcare-11-00320-f002:**
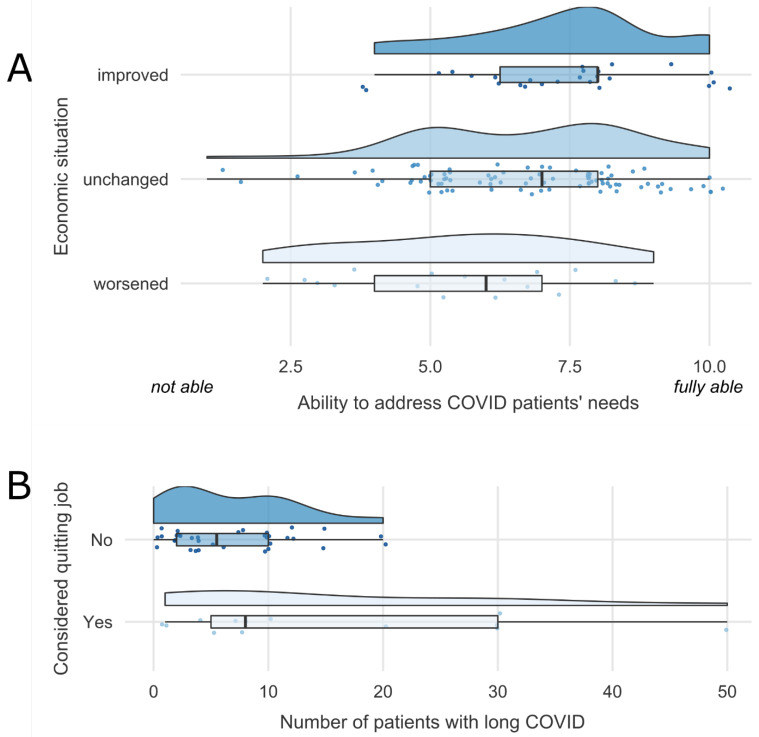
Influence of provision of care to patients with acute COVID-19 or long COVID on working conditions in GP practices. (**A**) Baseline 2021: GPs who reported a worsening of the economic situation of their practice since the outbreak of the pandemic rated their ability to address COVID patients’ needs as lower. (**B**) Follow-up 2022: GPs who considered quitting their job in the last 12 months currently treated more patients with long COVID.

**Figure 3 healthcare-11-00320-f003:**
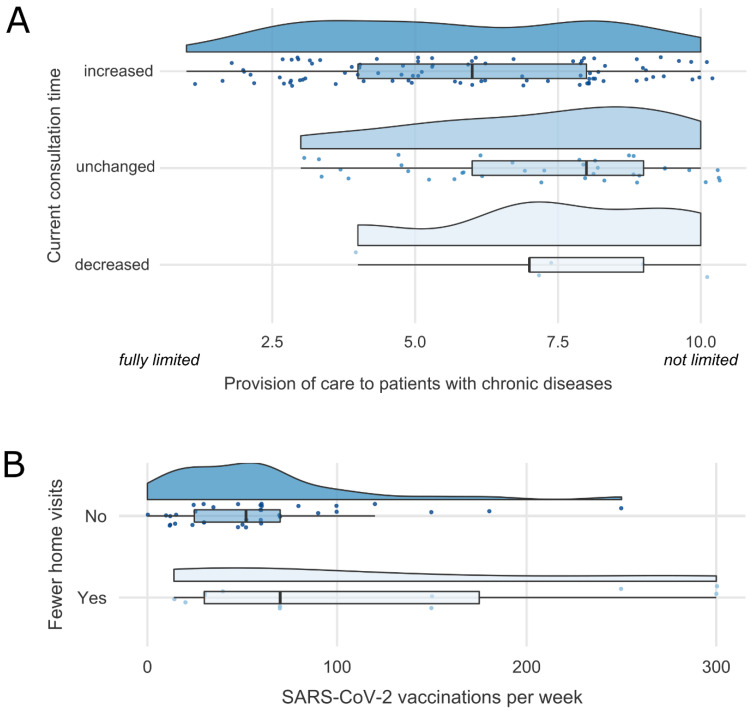
Influence of the pandemic on provision of care to other patient groups. (**A**) Baseline 2021: GPs who reported an increase in consultation time in their practice felt more limited in the satisfactory provision of medical care to other patients with chronic diseases compared to GPs who reported an unchanged or decreased consultation time. (**B**) Follow-up 2022: GPs who reported being able to offer fewer home visits than usual administered more SARS-CoV-2 vaccinations per week.

**Table 1 healthcare-11-00320-t001:** Baseline characteristics of participating GPs and their practices.

	Total	Practices with no Focus on COVID-19	Practices with Focus on COVID-19		
				*p*	F/*χ*2
*n*	143	132	11		
Age	50.2 ± 9.4	50.3 ± 9.2	48.5 ± 11.4	0.528	0.399
Sex	61.1% ♀	59.8% ♀	63.6% ♀	1.000	0.131
Medical specialist for *					
General Medicine	65%	66.7%	45.5%	0.193	2.009
Internal Medicine	33.6%	32.6%	45.5%	0.508	0.755
Others	4.9%	4.5%	9.1%	0.436	0.451
Practice structure				0.051	9.153
Single practice	61.5%	62.1%	54.5%		
Practice sharing	10.5%	11.4%	0.0%		
Joint practice	16.1%	13.6%	45.5%		
Medical center	11.9%	12.9%	0.0%		
Number of health insurance approved physicians per practice	1.5 ± 1.0	1.4 ± 0.9	2.2 ± 1.7	0.018	5.743
Number of cases per quarter				0.102	7.713
≤700	7.7%	8.3%	0.0%		
701–1000	24.5%	26.5%	0.0%		
1001–1500	39.2%	38.6%	45.5%		
≥1501	26.6%	24.2%	54.5%		
No answer	2.1%	2.3%	0.0%		
Catchment area of the practice *					
City	39.2%	38.6%	45.5%	0.751	0.198
Small town	44.8%	45.5%	36.4%	0.755	0.339
Rural	41.3%	41.7%	36.4%	1.000	0.118

Data are presented as mean, standard deviations, and percentage (n/n_valid_). *, multiple responses possible.

**Table 2 healthcare-11-00320-t002:** Influence of the pandemic on practice workload and differences between the two time points.

	Baseline 2021	Follow-Up 2022
*n*	143	51
Compared to pre-pandemic times, my practice is visited by patients		
More frequently	47.3%	69.4%
As frequently	42%	24.5%
Less frequently	10.7%	6.1%
Compared to pre-pandemic times, I need for patient consultations		
More time	67.9%	61.2%
As much time	28.2%	32.7%
Less time	3.8%	6.1%
Compared to pre-pandemic times, my workload is		
Higher	91.6%	95.9%
Just the same	8.4%	4.1%
Lower	0%	0%
Compared to pre-pandemic times, the economic situation of my practice		
Improved	19.8%	44.9%
Did not change	67.2%	40.8%
Worsened	13%	14.3%
Have you considered quitting your job in the last 12 months?	n.a.	
Yes		26.5%
No		73.5%

*Note*. Data are presented as percentage (n/nvalid). n.a., not assessed.

**Table 3 healthcare-11-00320-t003:** Consultations related to COVID-19/long COVID in GP practices.

	Baseline 2021	Follow-Up 2022
*n*	143	51
Current number of patients with acute COVID-19 per GP practice	n.a.	43.3 ± 84.2
Current number of patients with long COVID (symptoms lasting between 4–12 weeks) per GP practice	11.9 ± 11.3	10.0 ± 11.3
Current number of patients with long COVID (symptoms lasting >12 weeks) per GP practice	5.9 ± 6.4	6.8 ± 9.6
Consultation of patient with acute COVID-19 in practice		n.a.
Daily	32.1%	
Weekly	51.9%	
Monthly	13%	
Less than monthly	3%	
Never	0%	
Consultation of patient with long COVID (symptoms lasting between 4–12 weeks) in practice		n.a.
Daily	3.8%	
Weekly	44.3%	
Monthly	38.9%	
Less than monthly	13%	
Never	0%	
Consultation of patient with long COVID (symptoms lasting more than 12 weeks) in practice		n.a.
Daily	2.4%	
Weekly	16.9%	
Monthly	44.6%	
Less than monthly	24.6%	
Never	11.5%	
How many out of 100 patients are currently visiting for the following reasons:	n.a.	
Acute COVID-19 infection		9.1 ± 8.3
Long COVID (4–12 weeks after diagnosis)		3.2 ± 3.4
Long COVID (>12 weeks after diagnosis)		1.8 ± 1.9
Other infections		9.3 ± 6.1
Other acute reasons		13.7 ± 7.5
Care of chronic diseases		33.9 ± 16.5
SARS-CoV-2 vaccinations		15.3 ± 10.1
Other vaccinations		5.6 ± 4.7
Other reasons		7.9 ± 10
Number of SARS-CoV-2 vaccinations per week	n.a.	74.5 ± 73.7

Data are presented as mean, standard deviations, and percentage (n/n_valid_). n.a., not assessed.

**Table 4 healthcare-11-00320-t004:** Provision of care, current problems and need for support related to COVID-19/long COVID in GP practices.

	Baseline 2021	Follow-Up 2022
*n*	143	51
Problems related to treatment of patients with long COVID *		n.a.
Diagnosis is difficult	42.1%	
Unspecific symptoms	78.2%	
Uncertainty regarding medications	70.7%	
Long course of disease	78.9%	
No guidelines available	56.4%	
Need for support related to COVID-19 *		n.a.
Specialized information on typical symptoms and their length	53.4%	
Exchange with colleagues on case reports and workshops	45.9%	
Online training	37.6%	
Diagnostic tools	30.8%	
Special therapies for patients with long COVID	55.6%	
Guidelines on long COVID	63.9%	
Referral options to specialized outpatient clinics	60.9%	
Fewer appointments can be currently offered in the following services *:	n.a.	
Acute consultations		20%
Care of chronically ill patients		40%
Preventive cancer screenings		44%
Preventive health check-ups		58%
Self-payer services		66%
Home visits		24%
Nursing home visits		22%
Being able to address medical and/or psychological needs of COVID patients during consultations 1 = “not able”, 10 = “fully able”	6.7 ± 2	6.7 ± 2
Perceived limited provision of satisfactory medical care to other patients with chronic diseases 1 = “fully limited”, 10 = “not limited”	6.2 ± 2.5	5.1 ± 2.4

Data are presented as mean, standard deviations, and percentage (n/n_valid_). *, multiple responses possible, n.a., not assessed.

## Data Availability

The data that support the findings of this study are available on request from the corresponding author.

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
