# Peer review of "COVID-19 Continues to Burden General Practitioners: Impact on Workload, Provision of Care, and Intention to Leave"

_healthcare, 2023, doi:10.3390/healthcare11030320_

Round 1

Reviewer 1 Report

Authors did not use validated scales; they correctly flag his in the limitation paragraph, but they should explain why the choose to do so.

It seems that their study reached about one tenth of the Saxonian GP, that is not a real big gouge; only one third of them took part in the second wave of the study. Given that, I think that it is too ambitious to present the study as longitudinal one. It would be better to present the results of the first administration of the questionnaire and then adding a paragraph where the main results of the second wave are presented. 

The authors should elaborate a bit more how their study, in spite of the limitation, might be of some more general interest.

The communication is fine, the language is clear

Reviewer 2 Report

The authors presented us with a very interesting research on a problem that has affected the whole world for the last two and a half years, and in this way presented more research related to the importance of consultations in medical practice. 

At the very beginning, authors are suggested to shorten the abstract, as well as the key words. They should contain the necessary elements of the research objective, main results and significance. Everything else will be presented in the manuscript. Key words are excessive. Reduce the number.

The introductory part is coherent, but there is a lack of an overview of the research, or to expand the introductory part to include comparisons with similar research. Also, in that part that would be added, the setting of hypotheses is suggested.

The methodology is comprehensive and very detailed, but also precisely and clearly divided into chapters for better understanding of the readers.

It is learned that the results chapter is divided into several sections. Clarity is achieved both tabularly and graphically. The authors used an adequate statistical methodology, and presented all the results very clearly and no additional changes are needed.

The discussion is very detailed and extensive, which contributes to the significance of the manuscript and a better understanding of the research, which is of great importance. The authors outlined the limiting circumstances as well as future implications, both practical and theoretical.

It is suggested to change the conclusion and expand the literature references. Given that the research is very interesting and has wide significance in the world of science and practical areas, they could probably cite similar research in the discussion, comparing it with their results. Also, the list of references must be expanded. This would certainly have been achieved if they had part of the literature review, and in the discussion, a comparative analysis with similar research.

Proofread the manuscript before publication. Suggestions are given in the text above, and the proposal is to revise the English language.

Reviewer 3 Report

Review report for article entitled “COVID-19 continues to burden general practitioners: Impact on workload, provision of care, and job satisfaction”

This paper reports an empirical study examining changes in General Practitioners’ (GPs) workload, job satisfaction, and several other related variables across two occasions during the later stages of the COVID-19 pandemic. Although the paper’s content might be of interest to the journal’s readership, I feel that some revisions are required. I shall outline below my comments and suggestions that could help the authors improve the expository quality of the paper.

1. As the authors state, the main purpose of this paper is to explore the impact of the later phases of the pandemic on GPs’ consultations, practices, workload, job satisfaction etc. I feel that this is not entirely accurate since any observed changes in these variables do not necessarily reflect the impact of COVID-19. Other factors may be driving these changes. We cannot be certain of the causal effect of COVID-19 on these variables. I think that this argument needs some reworking.

2. I am also wondering weather Winter 2022 (I assume that it refers to the period from (September 2022 to December 2022) is still considered a period when COVID-19 is still a problem? The European Union (considering the context of this study) has declared since April 2022 that COVID-19 is no longer an emergency situation and many European countries no longer consider COVID-19 an issue. The authors could write more to convince the readers.

3. The rationale for the specific variable selection needs to be explained a little bit more.

4. Are all analyses based on the sample size of 51 GPs?

5. How many questions/items did the questionnaire include?

6. How was the online survey administered? How was the survey distributed and where?

7. In lines 159-160, it is stated that “Of these participants, 51 completed the follow-up survey in January or February 2022”. This is obviously not possible given that the paper was submitted in December 2022.

8. Were the exact same participants linked in both waves? If so, how were they identified across waves?

9. I think it makes no sense to present data on questions/items that were not available/asked in the first wave of the survey given the focus on “changes”.

10. Does it make sense to compare the GPs’ characteristics across the two occasions? Is it safe to assume that characteristics, such as practice structure or number of cases, remain the same?

11. The sample sizes between summer 2021 and winter 2022 (see Table 2) are quite unbalanced. How can we be certain that these differences in percentages are not due to the less sample size? Is it accurate to present these proportions comparatively? Also, it would be informative to present a statistical test of these differences.

12.  For within-subjects/repeated measures ANOVAs, I would like to see tests of the sphericity and equality of variances hypotheses.

13. In table 3, is it COVID or COVID-19? Please harmonise a little bit.

14. The results need clarity. I see that there are results based on repeated measures and results based on between-subjects’ differences. Hence, the results’ section needs some restructuring to reflect this distinction.

15. In the job satisfaction section (lines 273-287), SDs should be reported per mean.

16. In line 248, a correlation coefficient is presented. Why select this statistic instead of sticking to the comparisons or the percentages as in the rest of the paper?

17. Do you have an estimate of inter-coder reliability to judge the accuracy of coding? Did each coder code the open-ended responses separately and discuss afterwards?

18. Intention to leave the profession is called “professional attrition” is usually attributed to job burnout. If the authors did not assess job satisfaction directly as they claim (line 444), is it scientifically valid to be arguing about job satisfaction?

19. I am also wondering whether it would be sensible to conduct principal component analysis or factor analysis to reduce the number of variables in more concrete dimensions, such as “perceived burdens” (line 462), given that the substantial number of comparison variables becomes confusing.

20. Could the authors devote a paragraph in the introduction to describe how the variables under study function in the German context? I am also thinking that a description is needed regarding the situation with COVID-19 in Germany during the periods under study.

I hope that the above comments would prove helpful.

Round 2

Reviewer 1 Report

The paper has been improved and now can be published

Author Response

We thank the reviewer for accepting our manuscript.

Reviewer 3 Report

I thank the authors for revising the paper. The arguments are flowing better now and the additional material contributed to clarity of the content. My apologies for the misunderstanding with the dates. A few very minor issues that should be considered prior to publication are the following.

1. The ethics statement paragraph should be removed. All ethics statements should be placed at the end of the document before the references according to the journal style.

2.  A short comment would be in order justifying the choice between Pearson’s and Fisher’s chi-square test.

3. Given the revisions made, please harmonize the term “job satisfaction” with “intention to leave the profession” (line 429).

4. A comment or two in the introductory section indicating the potential contribution of this study to extant knowledge base and GPs’ practice would be helpful. The authors could use/transfer some of the content in the implications section.

5. The authors need to be more critical of the findings of differences in the discussion section. Given that the effect sizes are particularly low, I would caution against any overinterpretation of the differences.

6. To be honest, I still find it too hard to follow through the results section. Would it be possible to transfer all non-significant differences in the main text to the supplementary materials to help lessen the burden on the reader? The statistics can still be presented in the tables, though.

I wish the authors swift publication!
